# Base editing in human cells with monomeric DddA-TALE fusion deaminases

Young Geun Mok[1,3], Ji Min Lee[1,2,3], Eugene Chung[1,2,3], Jaesuk Lee[1,2], Kayeong Lim [1], Sung-Ik Cho[1,2] & Jin-Soo Kim [1✉]

Inter-bacterial toxin DddA-derived cytosine base editors (DdCBEs) enable targeted C-to-T conversions in nuclear and organellar DNA. DddA$_{tox}$, the deaminase catalytic domain derived from _Burkholderia cenocepacia_, is split into two inactive halves to avoid its cytotoxicity in eukaryotic cells, when fused to transcription activator-like effector (TALE) DNA-binding proteins to make DdCBEs. As a result, DdCBEs function as pairs, which hampers gene delivery via viral vectors with a small cargo size. Here, we present non-toxic, full-length DddA$_{tox}$ variants to make monomeric DdCBEs (mDdCBEs), enabling mitochondrial DNA editing with high efficiencies of up to 50%, when transiently expressed in human cells. We demonstrate that mDdCBEs expressed via AAV in cultured human cells can achieve nearly homoplasmic C-to-T editing in mitochondrial DNA. Interestingly, mDdCBEs often produce mutation patterns different from those obtained with conventional dimeric DdCBEs. Furthermore, mDdCBEs allow base editing at sites for which only one TALE protein can be designed. We also show that transfection of mDdCBE-encoding mRNA, rather than plasmid, can reduce off-target editing in human mitochondrial DNA.

[1] Center for Genome Engineering, Institute for Basic Science, Daejeon, Republic of Korea. [2] Department of Chemistry, Seoul National University, Seoul, Republic of Korea. [3] These authors contributed equally: Young Geun Mok, Ji Min Lee, Eugene Chung. ✉email: jskim01@snu.ac.kr

Programmable deaminases, which include CRISPR RNA-guided base editors[1–3], DddA-derived cytosine base editors (DdCBEs)[4], and zinc finger deaminases (ZFDs)[5], are broadly useful for targeted single-nucleotide substitutions in nuclear and organellar DNA in mammalian[4–7] and plant cells[8–10], paving the way for novel applications in medicine and biotechnology. Unlike programmable nucleases such as CRISPR-Cas9 and zinc finger nucleases, these site-specific deaminases do not produce DNA double-strand breaks in cells, avoiding unwanted mutations at on-target sites and undesired DNA damage responses via p53 activation. In particular, CRISPR RNA-free DdCBEs and ZFDs, composed of custom-designed transcription activator-like effector (TALE) arrays and zinc-finger DNA-binding proteins, respectively, fused to split interbacterial toxin DddA$_{tox}$ and a uracil glycosylase inhibitor (UGI), enable targeted C-to-T base conversions in mitochondrial DNA (mtDNA)[4–7] and chloroplast DNA[8–10], as well as nuclear DNA[5], in eukaryotic cells. CRISPR RNA-guided base editors are not suitable for organellar DNA editing owing to the difficulty of guide RNA delivery into organelles[11,12]. DddA$_{tox}$, an enzymatic moiety in the interbacterial toxin DddA derived from *Burkholderia cenocepacia*, catalyzes cytosine deamination within double-strand DNA (dsDNA) with a preference for cytosines found in a 5′-TC context. To avoid toxicity in host cells, DddA$_{tox}$ is split into two inactive halves[4–10], each of which is fused to custom-designed TALE arrays or zinc finger proteins (ZFPs) to make DdCBEs or ZFDs, respectively. A functional deaminase is reconstituted only when the two inactive halves are brought together on target DNA by two adjacently bound TALE proteins or ZFPs. C-to-T base conversions are induced in a spacer region of 14–18 base pairs (bp) or 7–15 bp in length between the two TALE-binding sites[4] or the two ZFP-binding sites[5], respectively.

Although DdCBEs and ZFDs are highly versatile, enabling targeted base editing in both nuclear and organelle DNA, the requirement for two TALE and ZFP constructs, respectively, rather than one, to induce base editing is disadvantageous and challenging. First, the use of two DNA-binding proteins limits targetable sites, because TALE proteins constructed with our golden-gate cloning kit preferentially bind to target DNA sites with a thymidine at both the 5′-and 3′-termini[13], although it is possible to design TALE proteins that recognize sites with no thymidine at the 5′[14], and because ZFPs prefer guanine-rich sites[15]. Second, the delivery of two constructs is often inefficient and challenging. Viral vectors with a limited capacity such as adeno-associated viral (AAV) vectors (capacity, ~4.7 kilobase pairs (kbps))[16], which are widely used in gene therapy, cannot accommodate a split DdCBE pair-encoding DNA sequence because the combination is too large (>5 kbps, including a promoter and a polyA signal). Furthermore, cloning two TALE array-encoding DNA sequences in a single vector with a larger capacity can also be difficult; such sequences are recombinogenic because of the high sequence homology within and between the two TALE arrays[17]. To overcome these limitations of dimeric systems employing split DddA$_{tox}$, here, we present non-toxic, full-length DddA$_{tox}$-fused DdCBEs, termed mDdCBEs (monomeric DdCBEs), for targeted C-to-T conversions in nuclear and organelle DNA.

## Results

### Rational design of non-toxic, full-length DddA$_{tox}$.
To obtain non-toxic, full-length DddA$_{tox}$ variants useful for base editing, we took two different approaches: structure-based, site-directed mutagenesis (Fig. 1) and random mutagenesis (Fig. 2). In the first approach, attenuated DddA$_{tox}$ variants with reduced affinity for DNA or with reduced catalytic activity were fused to catalytically dead CRISPR-Cas9 (dCas9) or nickase (nCas9) variants to create new base editors,

appropriate for targeted C-to-T conversions in human and other eukaryotic cells. To this end, we sought to subclone synthetic DNA segments encoding DddA$_{tox}$ variants, in which positively charged amino-acid residues were replaced with alanine (Fig. 1a), in an expression vector. We reasoned that these variants would bind to negatively charged dsDNA with reduced affinity, potentially avoiding toxicity. Most of the Ala-substituted variants failed to produce *E. coli* transformants (Fig. 1b). DNA sequencing of plasmids isolated from the resulting transformants showed that various frameshift mutations had been induced in the coding region. Apparently, these full-length DddA$_{tox}$ variants under the control of a mammalian promoter were weakly expressed in *E. coli*, leading to cell death. Fortunately, we were able to obtain several triple, quadruple, or quintuple (termed "AAAAA") Ala-substituted variants with no frameshift mutations. An active-site mutant, E1347A, was also successfully cloned.

We next investigated whether the AAAAA variant fused to D10A nCas9 (Cas9 nickase derived from *S. pyogenes*) or dCas9 (catalytically dead Cas9, containing D10A and H840A mutations, derived from S. pyogenes) and UGI could induce base editing in human embryonic kidney 293T (HEK293T) cells at a site in the *TYRO3* gene. Base editing frequencies were measured using targeted deep sequencing. Unlike Base Editor 2 (BE2) (or BE3), which is composed of the rat APOBEC1 deaminase, dCas9 (or D10A nCas9), and UGI and operates on a narrow window within the protospacer region, the AAAAA variant fusion-induced C-to-T conversions immediately upstream of the protospacer region at high frequencies of up to 43% (Fig. 1c–e). Unexpectedly, the E1347A variant fusion also induced base editing at the same C$_{-3}$ position with frequencies of 37% (nCas9 fusion) or 16% (dCas9 fusion) (Fig. 1c), which suggests that the E1347A mutation did not completely inactivate the deaminase activity of DddA$_{tox}$ and that the residual deaminase activity of the E1347A mutant was high enough to achieve base editing in human cells. Indeed, we were able to confirm the deaminase activity of the recombinant E1347A-D10A nCas9 fusion protein, expressed in and purified from *E. coli*, under cell-free in vitro conditions using a PCR amplicon containing the *TYRO3* site (Supplementary Fig. 1). The E1347A variant combined with the quintuple AAAAA mutation, however, failed to induce base editing. We also found that E1347A, AAAAA, and other Ala-substituted variants with no frameshift mutations, fused to dCas9 or nCas9 and UGI, were able to achieve base editing at positions up to 25 bp upstream of the protospacer region with frequencies of up to 26% at several other sites (Supplementary Figs. 2–4). Furthermore, these novel base editors were highly efficient in HeLa cells, exhibiting editing frequencies of up to 52% (Supplementary Fig. 5).

Importantly, base edits induced by these fusion proteins were maintained in cells for at least 21 days, demonstrating that these base editors were not cytotoxic (Supplementary Fig. 6). In fact, we were able to isolate clonal populations of cells with targeted edits at the *TYRO3* site and at *ROR1* site 1 induced by the AAAAA and E1347A fusion proteins. C-to-T edits were observed, with editing frequencies of up to 99%, at the *TYRO3* site and at the *ROR1* site 1 in 9 out of 11 clones (=81%) and 8 out of 17 clones (=47%), respectively, which had been treated using the AAAAA fusion. In addition, base edits were also observed at the *TYRO3* site and at the *ROR1* site 1 in 16 of 21 clones (=76%) and 5 of 23 clones (=21%), respectively, which had been treated with the E1347A fusion (Supplementary Fig. 7). These results indicate that these variants fused to the N terminus of D10A nCas9 are not cytotoxic.

### Base editing with DddA$_{tox}$ variants obtained by random mutagenesis.
In an attempt to create cytosine base editors with altered editing windows, we first tried to fuse each of these Ala-substituted variants to the C terminus of H840A nCas9.

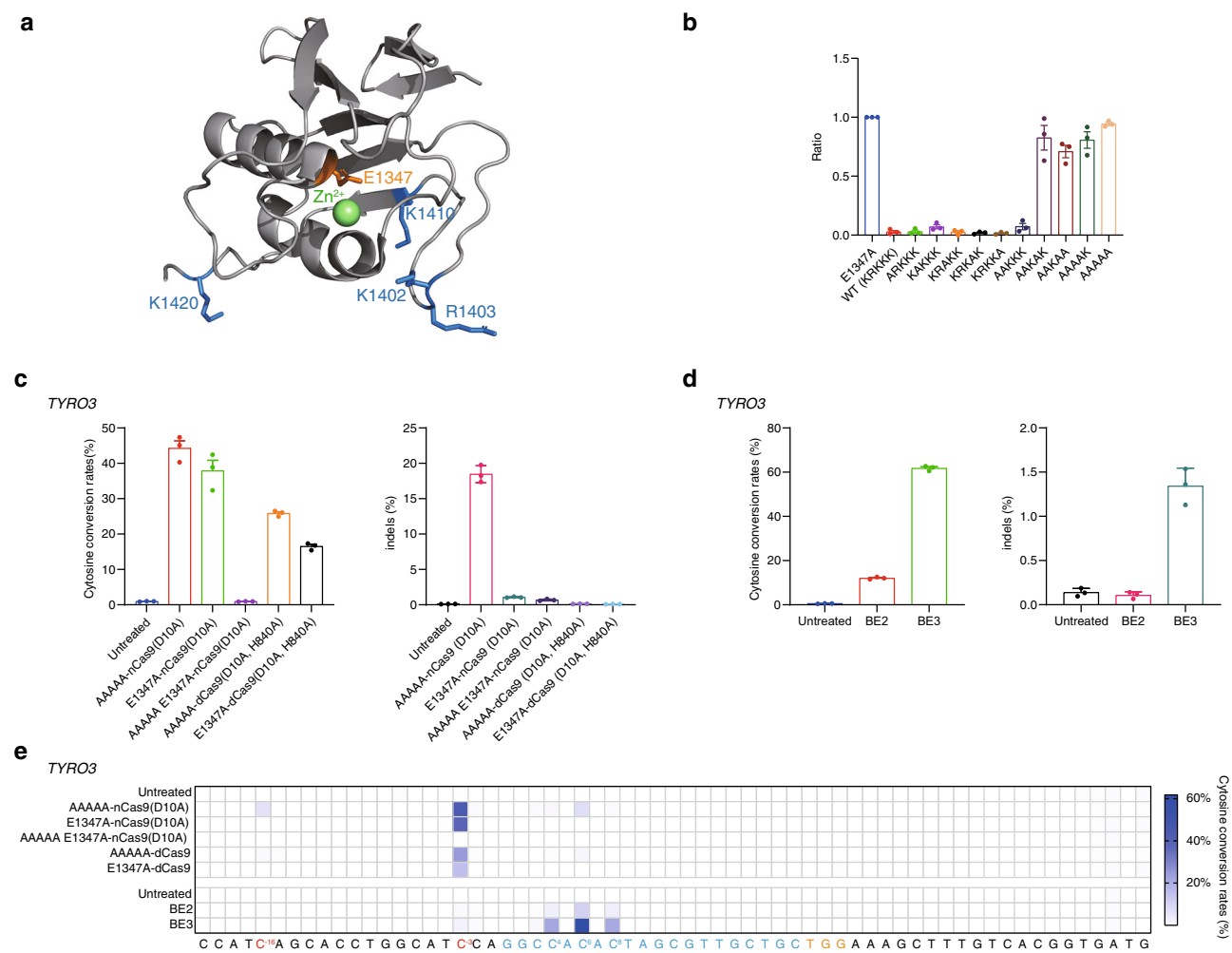

**Fig. 1 Characterization of DddA$_{tox}$ variants derived from structure-based screening. a** Crystal structure of DddA$_{tox}$ (gray, PDB 6U08). The positively charged amino-acid residues and E1347 active site are shown in blue and orange, respectively. K1424 is not shown because 1423–1427 residues are not resolved in the currently available crystal structure. green; Zn$^{2+}$ ion. **b** Graph showing the relative proportions of *E. coli* transformants obtained for each of the DddA$_{tox}$ variants containing alanine substitutions. The variant containing E1347A, a mutation affecting the active site, was used as a control. **c**, **d** Editing and indel frequencies induced by the indicated DddA$_{tox}$ AAAAA variant and E1347A variant fusions (**c**) and CBEs (**d**) at the *TYRO3* site. **e** Heat map showing the frequencies of C-to-T substitutions at various positions in the *TYRO3* site. The cytosine conversion rates and indel frequencies were measured by targeted deep sequencing. The protospacer is shown in blue and the PAM in orange. Cytosines upstream of the protospacer that underwent editing are shown in red. The numbers used to indicate the position of the DddA target window were obtained by counting backward from the proto-spacer, toward the 5′ upstream regions. Means ± s.e.m. (**b**–**d**) and heat map colors (**e**) were determined from three independent experiments. Source data are provided with this paper.

Unexpectedly, we failed to obtain intact constructs with no frameshift mutations. We, therefore, carried out error-prone PCR to introduce random mutations in the DddA$_{tox}$ coding sequence and were able to obtain a non-toxic, full-length DddA$_{tox}$ variant with four-point modifications: S1326G, G1348S, A1398V, S1418G (termed "GSVG"), after measuring base editing efficiencies of 23 clones with no frameshift mutations in human cells (Fig. 2a). This variant was also successfully fused to the C terminus or the N terminus of dCas9, D10A nCas9, H840A nCas9, and Cas9. In human cells, these fusion proteins, with the exception of those containing the wild-type Cas9, induced C-to-T conversions at various sites with efficiencies of up to 38% (Fig. 2b–d and Supplementary Figs. 8–11). Interestingly, fusion proteins containing the GSVG variant fused to the C terminus of dCas9, D10A nCas9, and H840A nCas9 showed cytosine base editing downstream of the protospacer adjacent motif (PAM), whereas those containing the same variant fused to the N terminus of dCas9 and nCas9 induced base editing upstream of the protospacer region (Fig. 2d).

As expected, fusion proteins containing Cas9 induced indels rather than base substitutions.

In order to pinpoint which mutations are critical in the GSVG variant, we attempted to create four revertants, namely, SSVG, GGVG, GSAG, and GSVS, via site-directed mutagenesis. We were able to obtain SSVG, GSAG, and GSVS revertants but failed to obtain the GGVG variant fused to the C terminus of nCas9. G1348 is right next to E1347, a key residue at the catalytic site. The G1348S mutation may reduce the catalytic activity, avoiding cytotoxicity in *E. coli*. We measured the editing frequencies induced by the three revertants and the GSVG variant at two target sites in transfected cells for up to 21 days. Frequencies of C-to-T edits induced by GSAG and GSVS gradually decreased by ~2-fold from day 3 to day 21 post-transfection, suggesting that these two revertants were somewhat cytotoxic, whereas those induced by GSVG and SSVG were maintained (Supplementary Fig. 12). These results suggest that G1348S in the GSVG variant is essential and that S1326G is neutral, whereas A1398V and S1418G reduce cytotoxicity.

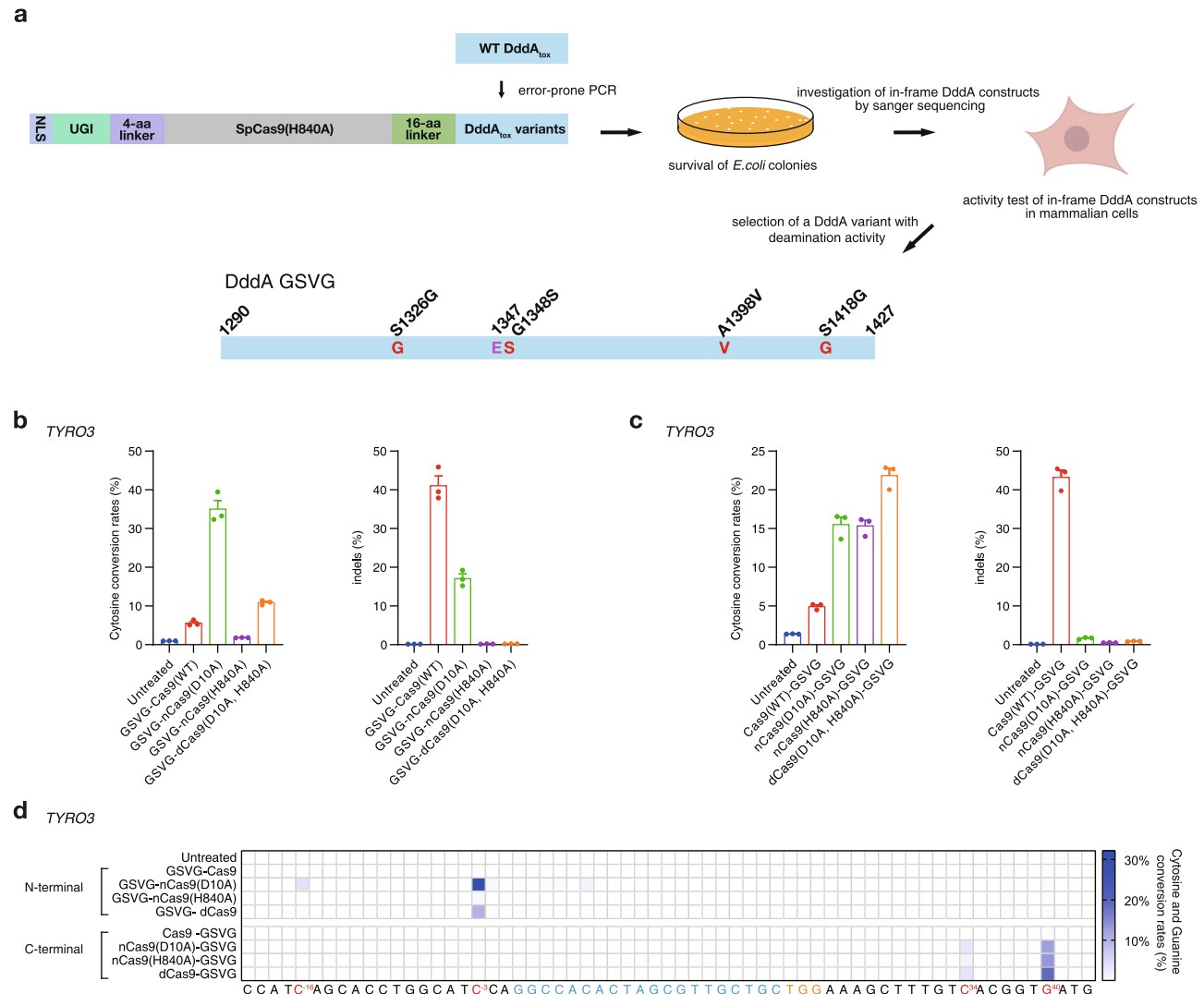

**Fig. 2 Characterization of the DddA_tox GSVG variant derived from random mutagenesis. a** Schematic diagram of the screen for non-toxic DddA_tox variants generated by error-prone PCR. The E1347 active site and replaced residues are shown in violet and red, respectively. **b, c** Editing and indel frequencies at the *TYRO3* site induced by the GSVG variant fused to the N (**b**) and C (**c**) termini of Cas9, D10A nCas9, H840A nCas9, and D10A, H840A dCas9 in HEK293T cells. **d**, Heat map showing the frequencies of C-to-T substitutions at various positions in the *TYRO3* site. The protospacer is shown in blue and the PAM in orange. Cytosines that underwent editing are shown in red. the N terminus and C terminus of Cas9, nCas9(D10A), nCas9(H840A), and dCas9, respectively. The numbers used to indicate the position of the GSVG target window were obtained by counting backward and forward from the proto-spacer, toward the 5' upstream and toward the 3' downstream regions, respectively. The base editing and indel frequencies were measured by targeted deep sequencing. Means ± s.e.m. (**b**, **c**) and heat map colors (**d**) were determined from three independent experiments. Source data are provided with this paper.

Taken together, our results show that non-toxic, full-length DddA_tox variants with reduced affinity for dsDNA (AAAAA), attenuated deaminase activity (E1347A and possibly GSVG), or reduced cytotoxicity (GSVG) can be fused to dCas9 or nCas9 to create novel base editors with altered editing windows. These base editors can be used for base editing at positions upstream or downstream of a protospacer region, which is beyond the reach of BE2 or BE3.

**mtDNA editing by monomeric DdCBEs**. We also investigated whether non-toxic, full-length DddA_tox variants could be used for mtDNA editing in human cells. Among several variants, only the GSVG and E1347A variants were successfully fused to the C terminus of TALE arrays designed to bind to three mitochondrial genes, *ND4*, *ND6*, and *ND1*. The resulting mDdCBEs containing the GSVG variant achieved base editing at intended target

nucleotide positions with high frequencies of up to 31% (*ND4*) (Fig. 3a, b), 27% (*ND6*) (Fig. 3c, d), and 42% (*ND1*) (Fig. 3e, f), on par with the original split DdCBE pairs targeted to these sites (shown as L-1397N + R-1397C, indicating DddA_tox split at G1397 fused to the left- or right-TALE array, and L-1333N + R-1333C, indicating DddA_tox split at G1333 fused to the left- or right-TALE array, in Fig. 3). mDdCBEs containing E1347A also generated targeted C-to-T conversions, albeit less efficiently, with frequencies of up to 7.2% (*ND4*), 8.9% (*ND6*), and 13.7% (*ND1*) at these sites. In addition, mDdCBEs were highly efficient in mouse NIH3T3 cells, exhibiting editing frequencies of up to 32% at a target site in the *MT-ND5* gene (Supplementary Fig. 13).

Interestingly, the split DdCBE pairs and mDdCBEs often produced different editing patterns. For example, the DdCBE pair (shown as L-1333N + R-1333C in Fig. 3b) specific to the *ND4* gene was poorly active at the C4 position with an editing

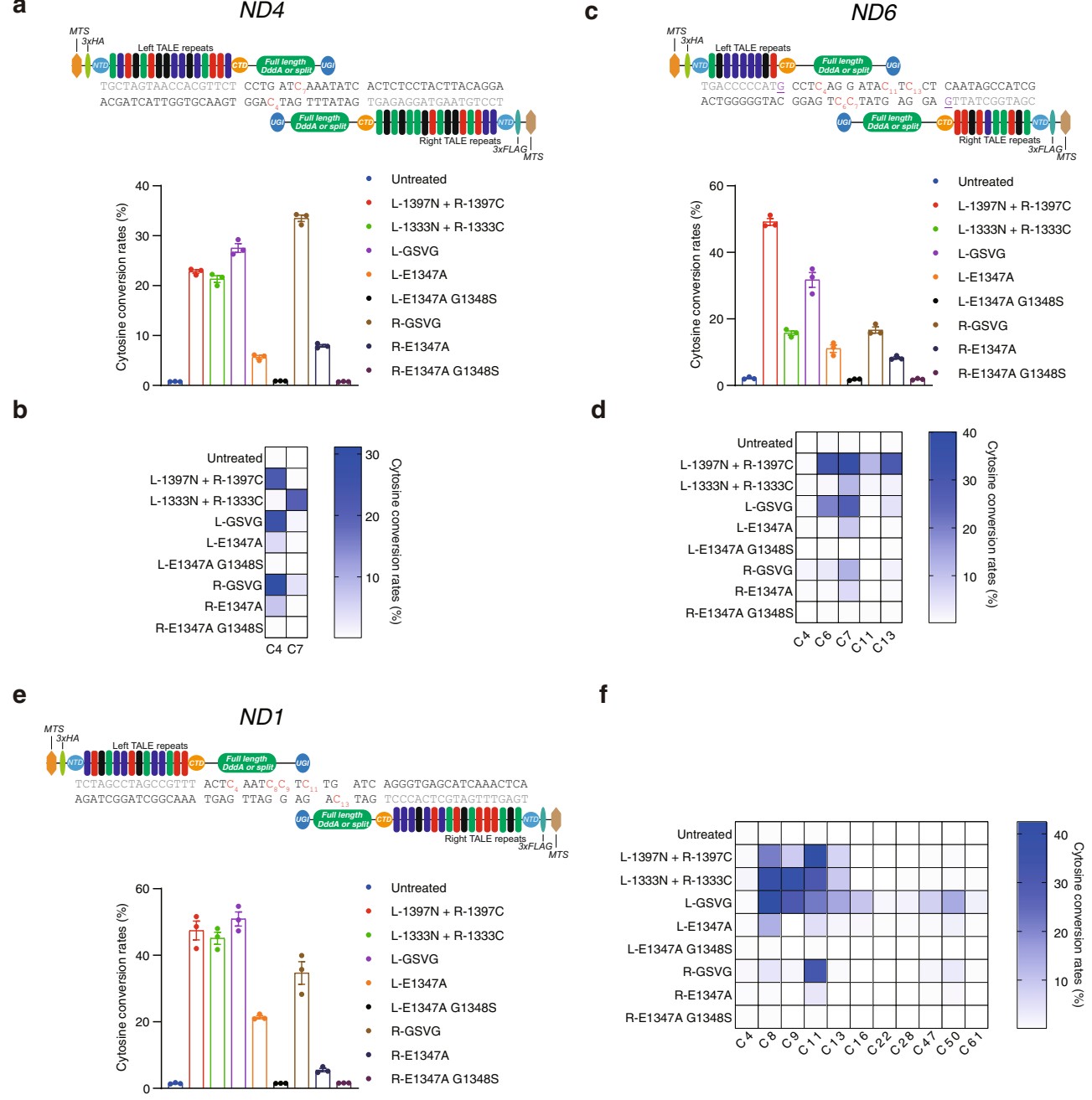

**Fig. 3 Base editing in the mitochondrial genome induced by mDdCBEs and DdCBEs. a**, **c**, **e** Editing efficiencies of mDdCBEs and DdCBEs at the *ND4* (**a**), *ND6* (**c**), and *ND1* (**e**) sites in HEK293T cells. Target cytosines and TALE-binding sites are shown red and gray, respectively. The left and right TALE arrays are represented by L and R, respectively. Mismatches between the site recognized by the *ND6*-specific TALE array and the reference genome are underlined and shown in purple. **b**, **d**, **f** Heat maps showing the frequencies of C-to-T conversions at the indicated positions in the *ND4* (**b**), *ND6* (**d**), and *ND1* (**f**) sites. The cytosine conversion rates were measured by targeted deep sequencing. Means ± s.e.m. (**a**, **c**, **e**) and heat map colors (**b**, **d**, **f**) were determined from three independent experiments. Source data are provided with this paper.

frequency of 0.8%, whereas two mDdCBEs containing the GSVG variant (shown as L-GSVG (Left TALE fused to the GSVG variant) and R-GSVG (Right TALE fused to the GSVG variant) in Fig. 3b) were highly active at this position with editing frequencies of 26 and 31% (Fig. 3b). We also noted that the *ND1* site-specific R-GSVG was highly selective, inducing C-to-T conversions primarily at the C11 position, whereas two split DdCBE pairs targeted to the same site and L-GSVG were poorly discriminatory, inducing base edits at multiple positions in the editing window (Fig. 3f). These results show that dimeric DdCBEs

and mDdCBEs can create different mutation patterns and suggest that the two forms can be complementary with each other to induce diverse mutations at a given target site.

We next investigated whether mDdCBEs targeted to the mitochondria were cytotoxic or would induce off-target mutations in the nuclear genome. We found that mtDNA edits induced by *ND6*-specific mDdCBEs were maintained in HEK293T cells for at least 21 days (Supplementary Fig. 14), suggesting that mtDNA editing by mDdCBEs was well-tolerated. We also found that mDdCBEs containing a mitochondrial

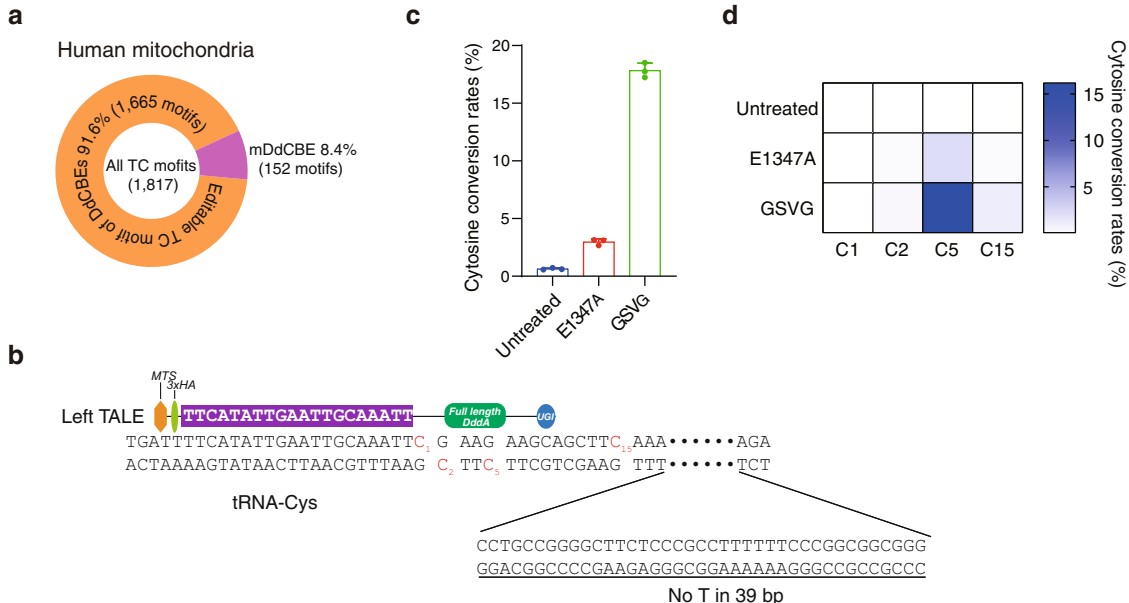

**Fig. 4 mDdCBE allows base editing in mtDNA at sites with a single TALE binding site. a** Among all TC motifs in human mtDNA, 8.4% can potentially be edited by mDdCBE but not DdCBE. **b** Schematic of a mDdCBE target site in the *MT-TC* gene in human mtDNA. Target cytosines are shown in red. **c** Editing frequencies induced by mDdCBEs at the *MT-TC* site in HEK293T cells. **d** Heat map showing the frequencies of C-to-T substitutions at various positions in the *MT-TC* site. The cytosine conversion rates were measured by targeted deep sequencing. Means ± s.e.m. (**c**) and colors in the heat map (**d**) were determined from three independent experiments. Source data are provided with this paper.

targeting sequence (MTS) rather than a nuclear localization signal did not induce off-target mutations at a potential off-target site with a single-nucleotide mismatch in the nuclear genome (Supplementary Fig. 15).

One important advantage of mDdCBEs over conventional dimeric DdCBEs is that mDdCBEs can be designed to target sites that contain only a single TALE-binding sequence (Fig. 4a), which typically have thymine at the 5′ termini. Although it is possible to ignore the need for thymine at the 5′ termini or to use an engineered TALE N-terminal domain that recognizes all four bases[14], it is unknown whether the resulting TALE proteins used in a DdCBE pair would be as efficient and specific as conventional TALE proteins recognizing thymine at the 5′ termini. As an example, we chose the *MT-TC* gene encoding tRNA-Cys: Various single-nucleotide substitutions in this gene are associated with myopathy or hearing loss[18]. We were able to design mDdCBEs but not dimeric DdCBEs to target a site in this gene, where there is no thymine within a stretch of 39 bp downstream of a potentially editable TC motif (Fig. 4b). The mDdCBE containing the GSVG variant targeted to this site achieved C-to-T conversions at a frequency of 16%, demonstrating the advantage of mDdCBEs over conventional DdCBEs (Fig. 4c, d).

Another advantage of mDdCBE over dimeric DdCBE is that the former can be delivered via a single AAV vector with a ~4.7-kb cargo capacity (Fig. 5a). We produced AAV2 particles encoding mDdCBEs targeted to the *ND4* and *ND1* sites and transduced HEK293T cells with variable viral doses. At day 6 post-infection, base editing frequencies reached as high as 99.1% at the *ND4* site and 59.8% at the *ND1* site with high multiplicity of infection (Fig. 5b–e). This result suggests that nearly homoplasmic (>99%) mutations can be induced in mtDNA using AAV-mediated DdCBE delivery.

We next examined the positions of C-to-T edits induced by mDdCBEs containing either the GSVG variant or the E1347A variant. We plotted the cytosine-editing frequencies of a total of 9 mDdCBEs at each nucleotide position downstream of a TALE-binding sequence to define an editing window for mDdCBEs

(Supplementary Fig. 16). Positions 4–11 were more frequently converted than those immediately adjacent to or far downstream of the TALE-binding site. Thus, the editing window for mDdCBEs can be loosely defined as spanning nucleotides 4–11 downstream of a TALE-binding site.

**Mitochondrial genome-wide target specificity.** We assessed the mitochondrial genome-wide target specificity of split DdCBEs and mDdCBEs by performing high-throughput sequencing of DNA samples isolated from cells transfected with the *ND1*- and *ND6*-specific split DdCBEs (shown as L-1333N + R-1333C (DddA$_{tox}$ split at G1333) and L-1397N + R-1397C (DddA$_{tox}$ split at G1397) in Fig. 6a, b) or mDdCBEs containing the E1347A or GSVG variant. In parallel, we also analyzed four TALE-free split or full-length DddA$_{tox}$ constructs to test whether they would induce random mutations across the mitochondrial genome. As expected, none of these TALE-free DddA$_{tox}$ constructs were mutagenic, compared with the negative control (no treatment): Average frequencies of mitochondrial genome-wide C-to-T editing induced by these constructs ranged from 0.018% to 0.019%, not much different from that obtained with the negative control (0.019%). The split DdCBE pairs targeted to the two sites, however, showed off-target C-to-T editing with average frequencies that ranged from 0.031% to 0.039% (*ND1*) or 0.14% to 0.19% (*ND6*). The mDdCBEs targeted to the same sites also were mutagenic, inducing off-target C-to-T editing with average frequencies that ranged from 0.041% to 0.21% (*ND1*) or 0.12% to 0.23% (*ND6*) (Fig. 6a, b). This result suggests that mDdCBE target specificities are not necessarily poorer than those of split DdCBE pairs.

Off-target activities of DdCBEs and ZFDs can be reduced or eliminated by delivering such constructs as in vitro transcripts rather than plasmids encoding them[5]. We investigated whether in vitro transcripts encoding mDdCBEs can also reduce off-target editing frequencies. We chose the two promiscuous *ND6*-targeted mDdCBEs, associated with relatively high average frequencies

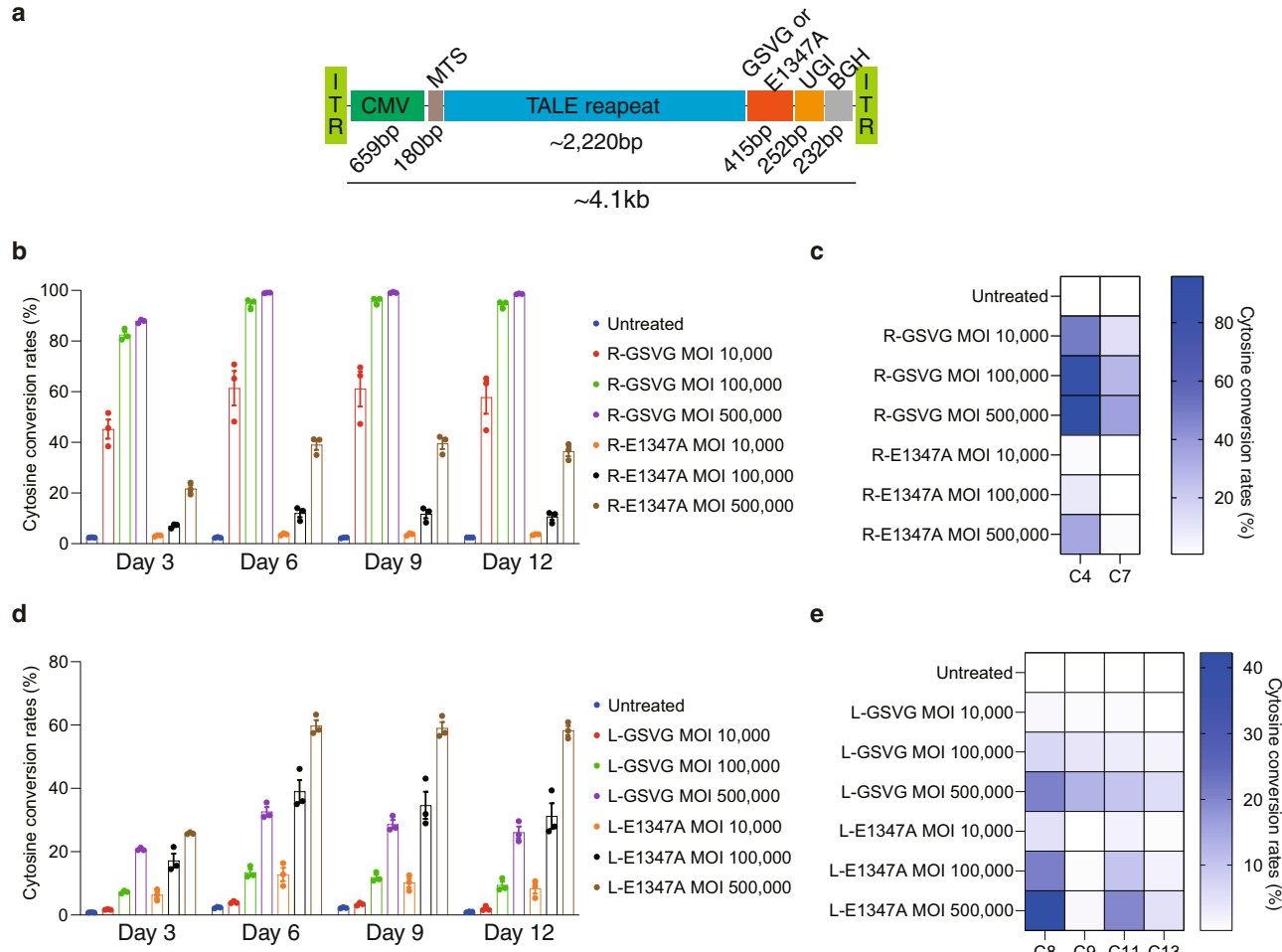

**Fig. 5 AAV-mediated base editing in mtDNA in HEK293T cells. a** Schematic showing the AAV vector encoding mDdCBE. Time-dependence of editing frequencies induced by AAV2-encoded mDdCBEs at the *ND4* (**b**) and *ND1* (**d**) sites. Heat maps showing the frequencies of C-to-T substitutions at various positions in the *ND4* (**c**) and *ND1* (**e**) sites on Day 12. The cytosine conversion rates were measured by targeted deep sequencing. Means ± s.e.m. (**b**, **d**) and colors in the heat maps (**c**, **e**) were determined from three independent experiments. Source data are provided with this paper.

of mitochondrial genome-wide off-target C-to-T editing, and transfected variable amounts of mDdCBE-encoding mRNA into HEK293T cells. As expected, high editing frequencies of up to 20% were obtained with increasing doses of up to 800 ng mRNA (Fig. 6c). Importantly, the use of 800 ng mRNA was as efficient as transfection of plasmid DNA in terms of on-target editing frequencies and reduced off-target C-to-T editing by 3.7 fold. Thus, the average frequency of mtDNA-wide off-target editing observed with 800 ng mRNA was 0.058% (Fig. 6d), whereas that obtained with the mDdCBE plasmid was 0.21% (Fig. 6b). Use of 200 ng mRNA achieved base editing with modest frequencies of up to 10% without inducing mtDNA-wide off-target editing, compared with that in the untreated control (Fig. 6c, d). These results show that the amount of mDdCBE-encoding mRNA can be titrated to reduce or avoid off-target editing in mtDNA.

## Discussion

In this study, we presented full-length, non-toxic DddA$_{tox}$ variants, which can be fused to D10A or H840A nCas9 to make novel CRISPR RNA-guided cytosine base editors with altered editing windows or to custom-designed TALE DNA-binding proteins to create mDdCBEs enabling mtDNA editing in human and mouse cells. mDdCBE-encoding genes, unlike those encoding dimeric DdCBEs, can be packaged into AAV and other viral vectors with limited cargo space, facilitating in vivo studies and gene therapy. Importantly, the *ND4*-specific mDdCBE delivered via AAV induced C-to-T edits at a frequency of up to 99.1%, suggesting that homoplasmic mutations can be achieved in mtDNA without drug selection. Furthermore, mDdCBEs can edit sites for which only a single TALE protein can be designed. We also found that mDdCBEs often yield mutation patterns different from those obtained with dimeric DdCBEs. We found, however, that certain, but not all, mDdCBEs were more prone to induce off-target editing in mtDNA, compared with dimeric DdCBEs. Fortunately, we were able to reduce or eliminate mitochondrial genome-wide off-target effects by using mDdCBE-encoding mRNAs rather than plasmid DNA. We believe that, together with dimeric DdCBEs and ZFDs for C-to-T editing and tran-scriptional activator-like effector nucleases (TALEDs) for A-to-G editing[19], mDdCBEs can broaden the scope of organellar genome editing.

## Methods

**Plasmid construction.** Sequences encoding the DddA$_{tox}$ variants were PCR amplified using the synthesized full-length DddA$_{tox}$-encoding sequence (gBlock, IDT) as a template, the primers listed in Supplementary Table 1, and Q5 DNA polymerase (NEB). The resulting PCR products were cloned using Gibson assembly (NEB) into p3s-BE3 that had been digested with BamH I and Sma I (NEB) in the *Apobec1* sequence. The TALE-DddA$_{tox}$ (Addgene #158093, #158095, #157842, #157841) plasmids were digested with BamH I and Sma I, and sequences

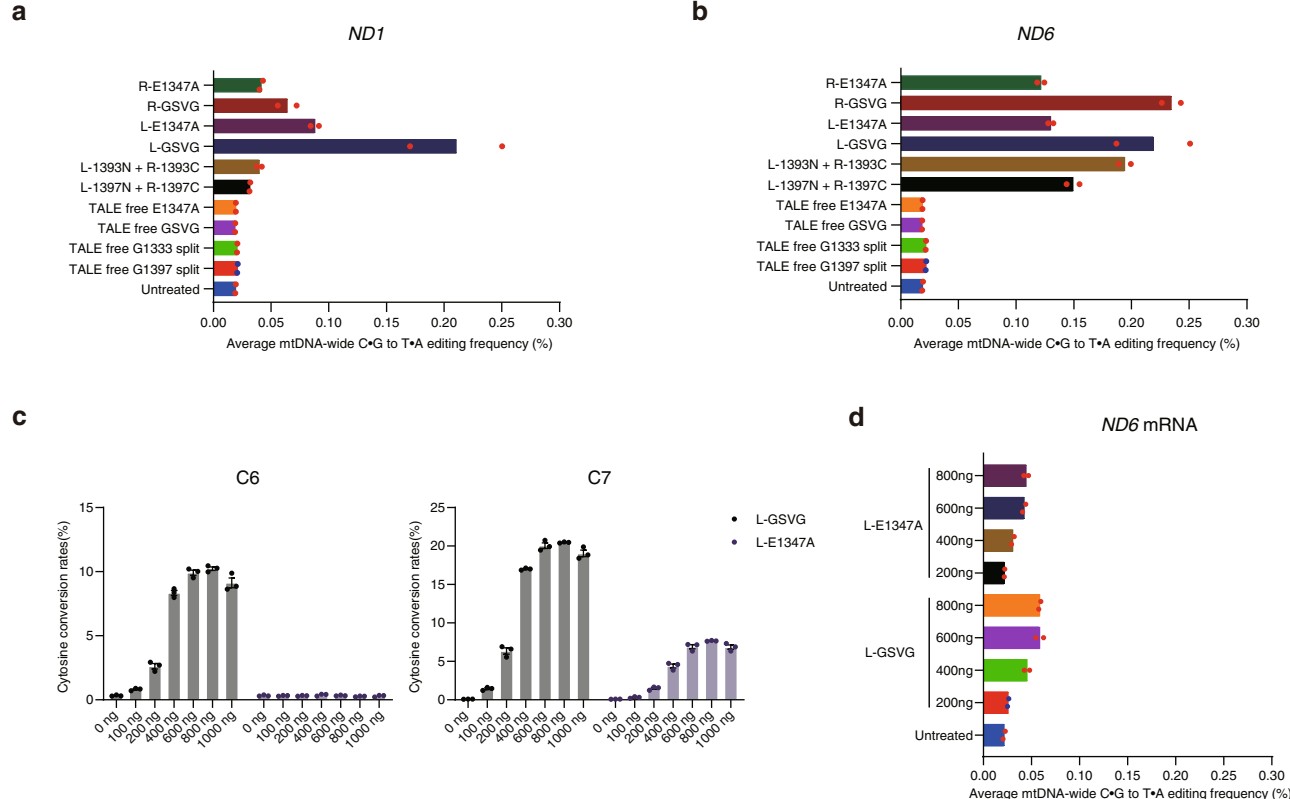

**Fig. 6 Mitochondrial genome-wide analysis of off-target editing by DdCBEs and mDdCBEs. a**, **b** Average frequencies of off-target C•G-to-T•A editing in mtDNA. HEK293T cells were transfected with plasmids encoding DdCBEs, mDdCBEs, TALE-free MTS-split DddA$_{tox}$-UGI, and TALE-free MTS-non-split DddA–UGI targeted to the *ND1* (**a**) and *ND6* (**b**) sites. **c** Editing frequencies at the C6 and C7 positions in the *ND6* site following transfection of various concentrations of mDdCBE-encoding mRNAs into HEK293T cells. **d** Average frequencies of off-target C•G-to-T•A editing in mtDNA from HEK293T cells transfected with mDdCBE-encoding mRNA. The cytosine conversion rates were measured by targeted deep sequencing. Means ± s.e.m. were determined from three (**c**) independent experiments. **a**, **b**, **d** Data are shown as the mean from two independent experiments. Source data are provided with this paper.

encoding the DddA$_{tox}$ variants were PCR amplified using the primers listed in Supplementary Table 1 and cloned into the digested plasmid using Gibson assembly (NEB). TALE arrays were designed to target human *ND1*, the human *MT-TC* gene encoding tRNA-Cys, and mouse *ND5* following the approach used in previous reports[6,8,13]. Assembled plasmids were chemically transformed into *E. coli* DH5α, and plasmids from the surviving colonies were analyzed by the Sanger sequencing method. Final plasmids were midi-prepped (Macherey-Nagel) for cell transfection.

To generate an AAV vector encoding mDdCBEs, PCR amplicons containing ND1 L-GSVG, ND1 L-E1347A, and the CMV promoter were assembly with Not I- (NEB) and Apa I- (NEB) digested pAAV vector by Gibson assembly (NEB). pAAV-CMV-ND1 L-GSVG was digested with EocR I (NEB) and Apa I (NEB) and ND4 R-GSVG, ND4 R-E1347A, ND6 L-GSVG, and ND6 L-E1347A were amplified by PCR for Gibson assembly (NEB). Assembled plasmids were chemically transformed into *E. coli* DH5α, and plasmids from the surviving colonies were analyzed by the Sanger-sequencing method. Final plasmids were midi-prepped (Macherey-Nagel) for viral particle production. The primers for PCR application are listed in Supplementary Table 1.

**Mammalian cell culture and transfection**. HEK293T (ATCC, CRL-11268) cells, HeLa (ATCC, CCL-2) cells, and NIH3T3 (ATCC, CRL-1658) cells were cultured and maintained at 37 °C in a 5% CO$_2$ atmosphere. Cells were grown in Dulbecco's Modified Eagle Medium supplemented with 10% (v/v) fetal bovine serum (Welgene) and 1% penicillin/streptomycin (Welgene). Twenty-four hours before transfection, cells were seeded in a 48-well plate (Corning) at a density of $8 \times 10^4$ cells/well (HEK293T) and $4 \times 10^4$ cells/well (HeLa), after which they were transfected with a plasmid encoding the Cas9-DddA$_{tox}$ fusion (750 ng) and a single guide RNA- (sgRNA-) encoding plasmid (250 ng) with Lipofectamine 2000 (Invitrogen) according to the manufacturer's protocol. NIH3T3 cells were seeded in a 12-well plate at a density of $1.5 \times 10^4$ cells/well before transfection. TALE-DddA$_{tox}$ constructs (200 ng) were transfected into HEK293T cells using Lipofectamine 2000. The sgRNA sequences are shown in Supplementary Table 2.

**Random mutagenesis**. Error-prone PCR was performed on the synthesized full-length DddA$_{tox}$-encoding sequence (gBlock, IDT) using a GeneMorph II Random

mutagenesis kit (Agilent) according to the manufacturer's protocol. In brief, 1, 100, and 700 ng of DddA$_{tox}$-encoding DNA fragments were used as template for the introduction of random mutations at a density of 0–16 mutations/kb. The full-length DddA$_{tox}$ gBlock sequence was PCR amplified beforehand with primers listed in Supplementary Table 1. PCR products were pooled and Gibson assembled (NEB) into p3s-UGI-Cas9(H840A)-DddA$_{tox}$(E1347A) that had been digested with Sma I(NEB) and Xho I (NEB). The assembled plasmids were transformed into DH5α heat-shock competent cells, after which plasmids from a fraction of the surviving colonies were subjected to Sanger sequencing. p3s-UGI-Cas9(H840A)-DddA$_{tox}$ plasmids with correct in-frame fusions were transfected into HEK293T cells along with a plasmid encoding an appropriate sgRNA to target the selected site. Editing activity was detected through targeted deep sequencing.

**Genomic and mtDNA preparation**. Cells transfected with plasmids encoding DddA$_{tox}$ variants fused to Cas9 were harvested 2 days post-transfection and cells transfected with plasmids encoding TALE-DddA$_{tox}$ were harvested 3 days post-transfection. Genomic and mtDNA were isolated using a DNeasy Blood & Tissue Kit (Qiagen). For large-scale analysis, DNA was extracted using 100 µL of cell lysis buffer (50 mM Tris–HCl, pH 8.0 (Sigma-Aldrich), 1 mM EDTA (Sigma-Aldrich), 0.005% sodium dodecyl sulfate (Sigma-Aldrich)) that included 5 µL of Proteinase K (Qiagen). The lysate was incubated at 55 °C for 1 h, and then at 95 °C for 10 min.

**Targeted deep sequencing**. To analyze the frequency of edits, on-target sites were amplified via nested primary PCR, a secondary PCR, and a third PCR using TruSeq HT Dual index-containing primers and PrimeSTAR® GXL DNA Polymerase (TAKARA) to generate deep sequencing libraries. The libraries were sequenced using Illumina MiniSeq with paired-end sequencing systems. The base editing and indel frequencies are presented as percentages of sequencing reads containing base conversions or indels among total sequencing reads. The computer program used to analyze the frequency of edits is available at https://github.com/ibs-cge/maund. The PCR primer sequences are shown in Supplementary Tables 3–5.

**In vitro transcription and transfection**. The templates for in vitro transcription were amplified by PCR using PrimeSTAR® GXL DNA Polymerase (TAKARA) with

the primers listed in Supplementary Table 1. mRNAs were synthesized from PCR amplicons using an mMESSAGE mMACHINE™ T7 ULTRA Transcription Kit (Invitrogen) according to the manufacturer's protocol. Various concentrations of the mRNAs were then transfected into cells with Lipofectamine 2000 (Invitrogen) and cells were harvested 3 days post-transfection for analysis.

**Analysis of mitochondrial genome-wide off-target effects**. mtDNA was amplified with PrimeSTAR® GXL DNA Polymerase (TAKARA) using the primers listed in Supplementary Table 6. Amplicons were purified with a Qiaquick PCR Purification Kit according to the manufacturer's protocol. Sequencing libraries were constructed using a Nextera DNA Flex Library Prep Kit (Illumina). Sequencing was performed with an Illumina MiniSeq with the 300 cycles paired-end program.

To analyze next-generation sequencing data from whole mitochondrial genome sequencing, we followed methods in previously published reports[4,5]. Fastq files were aligned to the GRCh38.p13 (release v102) reference genome using BWA (v.0.7.17), and BAM files were generated with SAMtools (v.1.9) by fixing read pairing information and flags. Positions with conversion rates ≥0.1% were identified among all cytosines and guanines in the mitochondrial genome using the REDItoolDenovo.py script from REDItools (v.1.2.1)[20]. In all samples, positions with ≥50% conversion and target sites considered as single nucleotide mutations in cell lines were excluded. To calculate the average C•G-to-T•A editing frequency for all C•Gs in the mitochondrial genome, the conversion rates were averaged at each base position in the off-target sites.

**Expression and purification of E1347A-nCas9(D10A) and in vitro activity assay**. For expression of DddA variant fragments with an N-terminal His purification tag in the pET vector, the constructs were transformed into the *E. coli* Rosetta2 (DE3) strain (Novagen). Transformants were grown at 37 °C in 4 L of LB medium supplemented with 50 µg/mL of kanamycin and 50 µg/mL of chloramphenicol until the OD600 reached 0.6. Protein expression was induced with 1 mM IPTG at 18 °C for 16 h. Cells were harvested by centrifugation at 6500 rpm for 15 min at 4 °C, and the pellet was resuspended in 100 mL of ice-cold column buffer (50 mM Tris–HCl pH 7.5, 500 mM NaCl, 30 mM imidazole, 1 mM DTT, 1 mg/mL lysozyme, 0.1 mM PMSF). The resuspended cells were subjected to three freeze–thaw cycles in liquid nitrogen and a water bath (37 °C), respectively, after which the cells were lysed by sonication for 9 min (5 s (on), 10 s (off)). The lysates were cleared by centrifugation at 15,000×*g* for 20 min. The supernatant was then incubated with nickel agarose beads (Ni-NTA; Qiagen) for 60 min with shaking at 4 °C. The lysate–resin mixture was loaded into a polypropylene column and washed two times with three column volumes of wash buffer (50 mM Tris–HCl pH 7.5, 500 mM NaCl, 30 mM imidazole, 1 mM DTT). Bound proteins were eluted with elution buffer (50 mM Tris–HCl pH 7.5, 500 mM NaCl, 300 mM imidazole, 1 mM DTT). The resulting protein fractions were concentrated with a centrifuge column (Amicon Ultra-4 Centrifugal Filter Devices; Millipore) at 6000×*g*.

For testing the deaminase activity of E1347A-nCas9(D10A) in vitro, a template was PCR-amplified from HEK293T genomic DNA using PrimeSTAR® GXL DNA Polymerase (TAKARA) with the *TYRO3* primers listed in Supplementary Table 3. A sgRNA targeting this site was synthesized by in vitro transcription using an mMESSAGE mMACHINE™ T7 ULTRA Transcription Kit (Invitrogen) according to the manufacturer's protocol. The PCR-amplified template (200 ng) and sgRNA (200 ng) were incubated with the purified protein (300 nM) in a 20 µL reaction containing NEB buffer 3.1 at 37 °C for various lengths of time ranging from 1 to 24 h. The reactions were terminated by incubation at 65 °C for 15 min. The templates were then PCR amplified for targeted deep sequencing.

**Transduction of AAV into HEK293T cells**. AAV serotype 2 was produced by the IBS Virus Facility (https://www.ibs.re.kr/virusfacility/). HEK293T cells were seeded at a density of 8 × 10⁴ cells/well in a 48-well plate (Corning) 24 h before transduction. They were then infected with different viral doses determined by quantitative PCR, and cultured in DMEM containing 10% FBS (Welgene). At different time points after transduction, cells were collected for targeted deep sequencing.

**In silico analysis of base-editable TC motifs in organelle DNA**. The human mitochondrial genome (NC_012920) was used as a reference sequence for this analysis. All TC motifs were extracted from the organelle DNA sequences, after which the split TALE-DddA$_{tox}$ recognition sites for each TC motif were determined according to the following criteria: (i) the TALE array-binding sequence must contain a thymidine at the 5′ and 3′ ends, (ii) the length of the TALE array-binding sequence must be 14–20 bp, and (iii) the sequence between a pair of TALE-binding sites must be 14–18 bp long, including at least one TC motif. After this analysis, the remaining TC motifs were analyzed to find locations at which only one TALE array could bind. For this procedure, criteria (i) and (ii) above were used; the third criterion was changed to (iii) TC motifs must be within 18 bp of the thymidine at the 3′ ends of a TALE array-binding sequence. The results of this analysis are shown in Supplementary Data 1.

**Reporting summary**. Further information on research design is available in the Nature Research Reporting Summary linked to this article.

## Data availability

The DNA sequencing data that support the findings of this study have been deposited in the NCBI's Sequence Read Archive (SRA) database with the accession code PRJNA 805019. The protein sequences of the newly synthesized TALE arrays and DddA$_{tox}$ variants are provided in the Supplementary Information. For data visualization, we used GraphPad Prism 8, Microsoft Excel 2016, PowerPoint 2016, and Adobe Illustrator CS6. The plasmid encoding DddA variants are available from Addgene (pCMV-AAAAA-nCas9(D10A)-UGI; #187409, pCMV-E1347A-nCas9(D10A)-UGI; #187410, pCMV-UGI-nCas9(H840A)-GSVG; #187411, pCMV-ND1 Left -GSVG-UGI; #187412, pCMV-ND1 Right-GSVG-UGI; #187413, pAAV-ND4 Right-GSVG-UGI; #187414, pAAV-ND4 Right-E1347A-UGI; #187415). Any other additional relevant data supporting the findings of this study are available from the authors upon reasonable request. Source data are provided with this paper.

## Code availability

Source code created by BotBot Inc. (https://github.com/ibs-cge/maund) was used to calculate base editing and indel frequencies from targeted deep sequencing data.

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

## Acknowledgements

This work was supported by the Institute for Basic Science (IBS-R021-D1 to J.-S.K.).

## Author contributions

Y.G.M., J.M.L., E.C., and J.-S.K. designed the study. Y.G.M., J.M.L., E.C., J.L., K.L., and S.-I.C. performed the experiments. Y.G.M. and J.-S.K. wrote the manuscript. J.-S.K. supervised the research.

## Competing interests

Y.G.M., J.M.L., E.C., K.L., S.-I.C., and J.-S.K. have submitted a provisional patent application based on results reported in this paper. J.-S.K. is a founder of and shareholder in ToolGen. All the other authors declare no competing interests.
