## [Peer Review File · Nature Communications]

Base editing in human cells with monomeric DddA-TALE fusion deaminasesREVIEWER COMMENTS

Reviewer #1 (Remarks to the Author):

The manuscript of Mok et al. describes the production of novel monomeric base editors. A publication in 2020 first described the cytosine deaminase DddA which uses double-stranded DNA as a substrate. This protein acts as a toxin in nature by inducing multiple mutations in target bacteria. Splitting DddA in two halves and fusing these halves to TALE DNA-binding domains prevented the toxic effect and allowed the establishment of dimeric TALE-base editors. This was a significant finding which expanded the classical CRISPR-based base editing systems and allowed editing the genomes of cellular organelles (mitochondria and chloroplasts).

In the present manuscript, the authors developed non-toxic mutant variants of DddA which show either a lower affinity for DNA or a lower deaminase activity and can thus be used as monomeric fusions to Cas9 or a single TALE DNA-binding domain. Such monomeric TALE-base editors were then used to edit nuclear and mitochondrial target sites with very high efficiency in human cells, demonstrating a very solid and overall convincing activity. The authors further demonstrate that the use of mRNA-encoded tools reduces off-target editing. Using Cas9-DddA fusions it was possible to edit sites that are outside of the targeting range of classical Cas9-base editors, demonstrating that the authors developed a very useful novel base editing tool.

Taken together, this is a highly interesting further development of Cas9-base editors and TALE-base editors. TALE-base editors have particular advantages over CRISPR/Cas-based systems, because they can edit the DNA of organelles. The use of monomeric tools makes this application much easier. Although I have a few critical comments, I believe that this work is highly useful and of broad general interest.

Major issues

the authors claim in the abstract that „...DddA-derived cytosine base editors (DdCBEs) function in pairs, which limits targetable sites and hampers gene delivery via viral vectors with a small cargo size.“ This statement is grossly over-selling the present data. No target sites have so far been published which can not be addressed using pairs of DddA base editors (or TALEN) unless the design is faulty. It is the particular strength of TALE-DNA binding domains that they can practically target any site. It is always possible to construct an artificial problem by using a very peculiar site and non-optimal TALE DNA-binding domains, but this is of no practical value. I suggest that the authors remove any sentence concerning targetable sites, because this is misleading.

Furthermore, the argument that two TALE-base editors don't fit into one viral vector is valid, but it remains unclear at this point why co-infection or co-transfection of the two parts of dimeric systems should not be possible. The argument in itself that a smaller tools is preferable is still valid, but again, the problem is not existential. In fact, the monomeric TALE-base editors themselves are larger than the individual units of the classical dimeric ones.

line 51-52: the statement that TALE proteins preferentially bind to target sites with a thymidine at both the 5' and 3'-termini" is false. Natural TALEs only require a T at the 5' end and even this requirement has been solved by different experimental means. Accordingly, this argument is wrong.

the authors used targeted deep sequencing to analyze the frequency of edits. Nevertheless, this is not mentioned in the text or the figure legends and the details are not given in the methods section. Please mention the experimental approach how editing frequencies were quantified directly in the main text (e.g. around line 86-90) and all figure legends. Furthermore, please specify (e.g. in the methods section) how many amplicons were pooled using barcoded primers and how many reads were analysed approximately per data point/experiment. Also, please add how long after transfection samples were taken for PCR amplification and whether the transfected cells were selected for or not.

Finally, please describe how the actual frequencies were calculated in the figure legends or methods section.

line 110: it remains slightly unclear how the GSVG mutant was obtained. How many different variants did the authors study? Figure 2 describes that a screen was conducted with probably multiple candidates being sequenced and tested in mammalian cells, before the GSVG variant was identified. This procedure is not explained in the main text at all and it remains unclear whether this variant was the only one with deamination activity.

line 173: the statement that TALEs "...must have a thymine at the 5' terminus." is wrong. Lamb et al. 2013 NAR 41, 9779-9785. describe a TALE N-terminal domain that recognizes all four bases, thereby eliminating the need for a T for the second TALE-base editor. Furthermore, there are other means, e.g. just to ignore a mismatch at the initial T, which can work as well. I find the paragraph in lines 171-179 highly misleading by constructing a non-relevant problem. Fig. 4a is completely speculative. In my opinion, the authors don't need this argument to support the manuscript, because constructing only one TALE-base editor, instead of two is clearly less work and this simplification will likely be highly welcome by the scientific community, even if dimeric TALE-base editors are possible for a given target site.

Minor issues

this is just a stylistic suggestion: the first sentence of the abstract contains more than 50 words. Such sentences are very difficult to read. If possible, I suggest to shorten this or split it into two sentences.

line 58-59: the authors state that TALE arrays are highly recombinogenic, because of high sequence homology within and between the two TALE arrays. This argument relies on a single publication (Holkers et al., 2013 NAR 41, e63) which used the lentiviral RNA polymerase for the application of TALEN. This polymerase is particularly error prone. Further observation of TALE repeat instabilities in other systems have not been published to my knowledge and dimeric TALE base editors have been very successfully applied (also by the authors) without report of particular instabilities. Although, recombinations in TALE repeats and between two TALEs happen in nature, this phenomenon doesn't appear to be a widespread problem for the application of TALE-based genome editing tools. Also, the risk of recombination within one repeat array is probably much larger than the risk of recombination between two TALE units. A monomeric system does not have any advantage in this regard. Therefore, the term „highly recombinogenic“ is too strong.

possibly a typing mistake: page 4, line 87 "APBEC1" might be "APOBEC1"

please specify in the figure legends which cell types are used. I believe that in Supplemental Figure 1-3 HEK293T cells are used and in Supplemental Figure 4, HeLa cells, but this is not indicated in the legend. Please also do this for other legends.

Fig. 2 might contain some spelling/ grammar errors. Instead of "survival E. coli colonies" I suggest to either use "surviving E. coli colonies" or "survival of E. coli colonies"; Also, instead of "selection the DddA variant with..." I suggest "selection of DddA variants (or a DddA variant)...".

Reviewer #2 (Remarks to the Author):

Mok et al. Base editing in human cells with monomeric DddA-TALE fusion deaminases

Because of the difficulty of guide RNA delivery into organelles, it is hard to edit mtDNA using CRISPR/Cas system. DddA-derived cytosine base editor (DdCBE) was produced by fusing DddAtox to TALE array, and can be used to edit the pathogenic mutation in mtDNA. DdCBE uses pairs of TALE array to make the functional DddAtox. This manuscript provided monomeric DddA-TALE fusion

deaminases. This monomeric DddA-TALE not only broaden the target sites, but also can be packaged into AAV and other viral vectors with a limited cargo space. The manuscript is appropriate for publication in Nature communications after some concerns being addressed.

Major concerns

1. The authors got the DddAtox variants by site-directed mutagenesis and random mutagenesis, and fused them to nCas9 to make novel base editors. The characteristic for these base editors were assessed using only one target site (TYRO3). More sites should be tested.
2. The authors concluded the DddAtox variants were not cytotoxic. The data from supplementary 5 showed the edited reads decreased as the days changed. The author should provide more evidences about the safety for the DddAtox variants.
3. One of the advantages for monomeric DddA-TALE is it can be packaged into AAV. The evidences about editing using AAV should be provided.

Minor concerns

1. The authors mentioned they got the DddAtox variants based on the structure. Figure 1a only showed the position of the amino acid. A crystal predicted structure of DddAtox to show the changed sites should be more convincing.
2. The detailed characteristic of the monomeric DddA-TALE should be summarized, like the editing window.
3. The use of 800 ng mRNA was efficient as transfection of plasmid DNA. How many cells do it transfect?

Reviewer #3 (Remarks to the Author):

In this study, the authors generated variants of DddAtox, the deaminase catalytic domains in the inter-bacterial toxin DddA from *Burkholderia cenocepacia*, by using structure-based site-directed mutagenesis and random mutagenesis. Some of these variants were fused to Cas9 nickase and TALE DNA-binding proteins for cytosine base editing in nuclear and mitochondrial DNA editing in human and mouse cells. The authors' approach enabled mitochondrial DNA editing using monomeric DdCBE, which is important progress in the field of mitochondrial genome editing. In addition, the authors also showed that mitochondrial genome-wide off-target effects can be reduced or eliminated by using mDdCBE-encoding mRNAs instead of plasmid DNA.

This well-designed study is timely. The authors achieved base editing using monomeric DdCBEs, which is an important contribution to the field of mitochondrial genome editing. I would recommend the publication of this exciting study after addressing only a minor point shown below.

Minor comment

1. The authors used E1347A for base editing, which is not compatible with the result of a previous study showing that the E1347A mutant is catalytically inactive (Mok et al. Nature 2020). Thus, this point should be addressed. I recommend testing the E1347A mutant under cell-free in vitro conditions whether it can induce cytosine deamination or not.

POINT-BY-POINT RESPONSE

We thank three anonymous reviewers for constructive criticisms and thoughtful comments. To address the critical issues raised by the reviewers, we have now included additional experimental data. We confirmed the catalytic activity of the E1347A DddA_{tox} mutant fused to the D10A Cas9 nickase under cell-free conditions using the recombinant protein. We also used AAV to express mDdCBEs in cultured human cells and found that a target site was edited with a frequency of up to 99.1%, suggesting that homoplasmic mutations can be induced in human mtDNA without drug selection. We believe that this is a major highlight in our revised manuscript and have added the following sentences in the abstract and the discussion section: **“We also show that mDdCBEs expressed via AAV in cultured human cells can achieve homoplasmic C-to-T editing in mitochondrial DNA.”** **“Importantly, the ND4-specific mDdCBE delivered via AAV induced C-to-T edits at a frequency of up to 99.1%, suggesting that homoplasmic mutations can be achieved in mtDNA without drug selection.”**

Reviewer #1 (Remarks to the Author):

Taken together, this is a highly interesting further development of Cas9-base editors and TALE-base editors. TALE-base editors have particular advantages over CRISPR/Cas-based systems, because they can edit the DNA of organelles. The use of monomeric tools makes this application much easier. Although I have a few critical comments, I believe that this work is highly useful and of broad general interest.

Major issues

the authors claim in the abstract that „...DddA-derived cytosine base editors (DdCBEs) function in pairs, which limits targetable sites and hampers gene delivery via viral vectors with a small cargo size.“ This statement is grossly over-selling the present data. No target sites have so far been published which can not be addressed using pairs of DddA base editors (or TALEN) unless the design is faulty. It is the particular strength of TALE-DNA binding domains that they can practically target any site. It is always possible to construct an artificial problem by using a very peculiar site and non-optimal TALE DNA-binding domains, but this is of no practical value. I suggest that the authors remove any sentence concerning targetable sites, because this is misleading.

Response: Although we respectively disagree with this reviewer, we understand his or her concern. We have now deleted the phrase, “limits targetable sites”, from the abstract.

Furthermore, the argument that two TALE-base editors don't fit into one viral vector is valid, but it remains unclear at this point why co-infection or co-transfection of the two parts of dimeric systems should not be possible. The argument in itself that a smaller tools is preferable is still valid, but again, the problem is not existential. In fact, the monomeric TALE-base editors themselves are larger than the individual units of the classical dimeric ones.

Response: Cloning two viral vectors and producing two recombinant viruses are more costly and laborious than cloning one vector and producing one recombinant virus. Furthermore, not every cell in a cell culture or in vivo will be co-infected by two recombinant viruses. Cells infected by a single virus will not be edited if two vectors are needed to deliver dimeric systems. Although monomeric DdCBEs themselves are larger than each monomer in a dimeric system, they can be fit into a single AAV vector, as shown in Fig. 5.

line 51-52: the statement that TALE proteins preferentially bind to target sites with a thymidine at both the 5' and 3'-termini“ is false. Natural TALEs only require a T at the 5' end and even this requirement has been solved by different experimental means. Accordingly, this argument is wrong.

Response: We have now added the following phrase in the relevant sentence: “although it is possible to design TALE proteins that recognize sites with no thymidine at the 5' or 3' terminus...”

the authors used targeted deep sequencing to analyze the frequency of edits. Nevertheless, this is not mentioned in the text or the figure legends and the details are not given in the methods section. Please mention the experimental approach how editing frequencies were quantified directly in the main text (e.g. around line 86-90) and all figure legends. Furthermore, please specify (e.g. in the methods section) how many amplicons were pooled using barcoded primers and how many reads were analysed approximately per data point/experiment. Also, please add how long after transfection samples were taken for PCR

amplification and whether the transfected cells were selected for or not. Finally, please describe how the actual frequencies were calculated in the figure legends or methods section.

Response: We have now mentioned in the main text and figure legends that base editing frequencies were measured using targeted deep sequencing and added the following sentence on p.4: “Base editing frequencies were measured using targeted deep sequencing.”

We have also mentioned in the Methods that “The base editing and indel frequencies are presented as percentages of sequencing reads containing base conversions or indels among total sequencing reads.”

The raw DNA sequencing data used for the base editing and indel analysis are provided as a source data file, with accession code PRJNA 805019 at NCBI.

Samples for targeted deep sequencing were previously described in the Methods section as follows: “Cells transfected with plasmids encoding DddA_{tox} variants fused to Cas9 were harvested at day 2 post-transfection and cells transfected with plasmids encoding TALE-DddA_{tox} were harvested at day 3 post-transfection”.

line 110: it remains slightly unclear how the GSVG mutant was obtained. How many different variants did the authors study? Figure 2 describes that a screen was conducted with probably multiple candidates being sequenced and tested in mammalian cells, before the GSVG variant was identified. This procedure is not explained in the main text at all and it remains unclear whether this variant was the only one with deamination activity.

Response: To explain how we obtained the GSVG variant in more detail, we have now added the following phrase (underlined) in the sentence on p.6, as follows: “We, therefore, carried out error-prone PCR to introduce random mutations in the DddA_{tox} coding sequence and were able to obtain a non-toxic, full-length DddA_{tox} variant with four point modifications: S1326G, G1348S, A1398V, S1418G (termed “GSVG”), after measuring base editing efficiencies of 23 clones with no frameshift mutations in human cells (Fig. 2a)”.

line 173: the statement that TALEs "...must have a thymine at the 5' terminus." is wrong. Lamb et al. 2013 NAR 41, 9779-9785. describe a TALE N-terminal domain that recognizes

all four bases, thereby eliminating the need for a T for the second TALE-base editor. Furthermore, there are other means, e.g. just to ignore a mismatch at the initial T, which can work as well. I find the paragraph in lines 171-179 highly misleading by constructing a non-relevant problem. Fig. 4a is completely speculative. ^[SEP]In my opinion, the authors don't need this argument to support the manuscript, because constructing only one TALE-base editor, instead of two is clearly less work and this simplification will likely be highly welcome by the scientific community, even if dimeric TALE-base editors are possible for a given target site.

Response: To address this reviewer's concern, we have now modified the sentence in question and added a sentence (underlined), as follows: “One important advantage of mDdCBEs over conventional dimeric DdCBEs is that mDdCBEs can be designed to target sites that contain only a single TALE-binding sequence (Fig. 4a), which typically have a thymine at the 5' terminus. Although it is possible to ignore the need for a thymine at the 5' terminus or to use an engineered TALE N-terminal domain that recognizes all four bases (Lamb et al. 2013), it is unknown whether the resulting TALE proteins used in a DdCBE pair would be as efficient and specific as conventional TALE proteins recognizing a thymine at the 5' terminus.”

We would like to keep Figure 4a, because it is informative and is an introduction to the rest of Figure 4, describing the advantage of mDdCBEs over dimeric DdCBEs. The other reviewers may also agree with us. We will consult with the editor whether we will need to delete Fig. 4a or all of Fig. 4.

Minor issues

this is just a stylistic suggestion: the first sentence of the abstract contains more than 50 words. Such sentences are very difficult to read. If possible, I suggest to shorten this or split it into two sentences.

Response: This is a good suggestion. We have now divided the sentence as follows: “Inter-bacterial toxin DddA-derived cytosine base editors (DdCBEs) enable targeted C-to-T conversions in nuclear and organellar DNA. DddA_{tox}, the deaminase catalytic domain derived from *Burkholderia cenocepacia*, is split into two inactive halves to avoid its cytotoxicity in eukaryotic cells, when fused to transcription activator-like effector (TALE) DNA-binding proteins to make DdCBEs.”

line 58-59: the authors state that TALE arrays are highly recombinogenic, because of high

sequence homology within and between the two TALE arrays. This argument relies on a single publication (Holkers et al., 2013 NAR 41, e63) which used the lentiviral RNA polymerase for the application of TALEN. This polymerase is particularly error prone. Further observation of TALE repeat instabilities in other systems have not been published to my knowledge and dimeric TALE base editors have been very successfully applied (also by the authors) without report of particular instabilities. Although, recombinations in TALE repeats and between two TALEs happen in nature, this phenomenon doesn't appear to be a widespread problem for the application of TALE-based genome editing tools. Also, the risk of recombination within one repeat array is probably much larger than the risk of recombination between two TALE units. A monomeric system does not have any advantage in this regard. Therefore, the term „highly recombinogenic“ is too strong.

Response: Please note that unsuccessful cloning is seldom reported in the literature. To address this reviewer's concern, we have now deleted "highly" from the sentence.

possibly a typing mistake: page 4, line 87 "APBEC1" might be "APOBEC1"

Response: We have now corrected it. We would like to thank this reviewer for careful reading.

please specify in the figure legends which cell types are used. I believe that in Supplemental Figure 1-3 HEK293T cells are used and in Supplemental Figure 4, HeLa cells, but this is not indicated in the legend. Please also do this for other legends.

Response: We have now indicated the cell lines used in the experiments described in the figure legends.

Fig. 2 might contain some spelling/ grammar errors. Instead of "survival E. coli colonies" I suggest to either use "surviving E. coli colonies" or "survival of E. coli colonies"; Also, instead of "selection the DddA variant with..." I suggest "selection of DddA variants (or a DddA variant)...".

Response: We would like to thank this reviewer again for helpful suggestions.

a

Reviewer #2 (Remarks to the Author):

Mok et al. Base editing in human cells with monomeric DddA-TALE fusion deaminases

Because of the difficulty of guide RNA delivery into organelles, it is hard to edit mtDNA using CRISPR/Cas system. DddA-derived cytosine base editor (DdCBE) was produced by fusing DddAtox to TALE array, and can be used to edit the pathogenic mutation in mtDNA. DdCBE uses pairs of TALE array to make the functional DddAtox. This manuscript provided monomeric DddA-TALE fusion deaminases. This monomeric DddA-TALE not only broaden the target sites, but also can be packaged into AAV and other viral vectors with a limited cargo space. The manuscript is appropriate for publication in Nature communications after some concerns being addressed.

Major concerns

1. The authors got the DddAtox variants by site-directed mutagenesis and random mutagenesis, and fused them to nCas9 to make novel base editors. The characteristic for these base editors were assessed using only one target site (TYRO3). More sites should be tested.

Response: Please note that the editing activity of DddA_{tox} variants obtained by site-directed mutagenesis was assessed at ROR1 site 1, ROR1 site 2, ROR1 site 3, FANCF, HBB, HEK3, EMX1 site 1, and TRAC5 site 1, in addition to TYRO3, as shown in Supplementary Figures 2-3. Furthermore, the activity of DddAtox variants obtained by random mutagenesis was assessed at EMX1 site 1, EMX1 site 2, ROR1 site 1, ROR1 site 2, ROR1 site 3, FANCF, HBB, and TRAC5 site 2, as shown in Supplementary Figures 8, 10.

Supplementary Figure 2

a

TYRO3

b

ROR1 Site 1

c

HEK3

Supplementary Figure 3

Supplementary Figure 8

a *EMX1* site 1

b *EMX1* site 2

c *ROR1* site 2

d *HBB*

Supplementary Figure 10

2.The authors concluded the DddAtox variants were not cytotoxic. The data from supplementary 5 showed the edited reads decreased as the days changed. The author should provide more evidences about the safety for the DddAtox variants.

Response: To address this issue, we have now obtained several clonal populations of cells derived from single cells, and described the results on p.5, as follows: **“In fact, we were able to isolate clonal populations of cells with targeted edits at the *TYRO3* site and at *ROR1* site 1 induced by the AAAAA and E1347A fusion proteins. C-to-T edits were observed, with editing frequencies of up to 99%, at the *TYRO3* site and at**

ROR1 site 1 in 9 out of 11 clones (= 81%) and 8 out of 17 clones (= 47%), respectively, which had been treated using the AAAAA fusion. In addition, base edits were also observed at the TYRO3 site and at ROR1 site 1 in 16 of 21 clones (= 76%) and 5 of 23 clones (= 21%), respectively, which had been treated with the E1347A fusion (Supplementary Fig. 7). These results indicate that these variants fused to the N terminus of D10A nCas9 are not cytotoxic.”

Supplementary Figure. 7

a AAAAA-D10A nCas9

TYRO3

	Editing frequency (%)		
	C-6	C3	C16
#1	0.13	0.23	0.08
#2	0.09	99.8	0.11
#3	0.12	99.6	47.1
#4	0.08	22.2	0.32
#5	0.05	40.8	38.9
#6	3.8	50.9	1.16
#7	51.5	99.7	47.5
#8	0.37	6.5	2.9
#9	0.1	0.18	0.12
#10	0.09	99.8	48.9
#11	50.8	99.7	0.07

ROR1 site 1

	Editing frequency (%)	
	C12	C25
#1	57.8	0.11
#2	0.09	0.06
#3	0.1	0.1
#4	0.14	0.14
#5	57.4	0.09
#6	0.07	0.08
#7	4.1	0.09
#8	0.06	0.13
#9	1.7	0.12
#10	27.5	0.19
#11	0.1	0.1
#12	0.11	0.11
#13	0.12	0.09
#14	5.1	19.4
#15	77.1	22.9
#16	0.07	0.07
#17	60.5	0.06

b E1347A-D10A nCas9

TYRO3

	Editing frequency (%)	
	C3	C16
#1	0.16	0.08
#2	26.4	0.08
#3	0.1	0.09
#4	50.1	0.11
#5	0.15	0.13
#6	51.8	0.07
#7	90.5	0.08
#8	0.12	0.12
#9	60.6	0.08
#10	99.4	0.13
#11	50.4	0.1
#12	50.5	0.12
#13	99.5	0.12
#14	77.7	0.11
#15	49.5	0.12
#16	48.4	0.11
#17	30	0.06
#18	49.4	0.08
#19	0.11	0.15
#20	1.75	0.09
#21	50.2	0.09

ROR1 site 1

	Editing frequency (%)	
	C12	C25
#1	0.06	0.12
#2	0.09	0.07
#3	1.11	0.15
#4	0.08	0.08
#5	0.09	0.13
#6	25.4	0.11
#7	0.07	0.15
#8	0.09	0.1
#9	0.06	0.09
#10	0.08	0.05
#11	0.08	0.09
#12	20.3	0.12
#13	0.04	0.13
#14	0.06	0.09
#15	20.9	0.12
#16	0.04	0.1
#17	0.06	0.1
#18	0.09	0.08
#19	0.09	0.08
#20	0.06	0.15
#21	0.05	0.11
#22	0.07	0.11
#23	21.1	0.11

3. One of the advantages for monomeric DddA-TALE is it can be packaged into AAV. The evidences about editing using AAV should be provided.

Response: We have now added Fig. 5 to show AAV-mediated editing frequencies. The results have now been described on p.9 as follows: “Another advantage of mDdCBE

over dimeric DdCBE is that the former can be delivered via a single AAV vector with a ~4.7-kb cargo capacity (Fig. 5a). We produced AAV2 particles encoding mDdCBEs targeted to the *ND4* and *ND1* sites and transduced HEK293T cells with variable viral doses. At day 6 post-infection, base editing frequencies reached as high as 99.1% at the *ND4* site and 59.8% at the *ND1* site with high multiplicity of infection (Fig. 5b-e). This result suggests that nearly homoplasmic (> 99%) mutations can be induced in mtDNA using AAV-mediated DdCBE delivery.”

Minor concerns

1. The authors mentioned they got the DddAtox variants based on the structure. Figure 1a only showed the position of the amino acid. A crystal predicted structure of DddAtox to show the changed sites should be more convincing.’

Response: This is a good suggestion. We have now added the 3D structure as Figure 1a, as shown below.

a

2. The detailed characteristic of the monomeric DddA-TALE should be summarized, like the editing window.

Response: We have now added the following paragraph on p.9: “We next examined the positions of C-to-T edits induced by mDdCBEs containing either the GSVG variant or the E1347A variant. We plotted the cytosine-editing frequencies of a total of 9 mDdCBEs at each nucleotide position downstream of a TALE-binding sequence to define an editing window for mDdCBEs (Supplementary Fig. 16). Positions 4 to 11 were more frequently converted than those immediately adjacent to or far downstream of the TALE-binding site. Thus, the editing window for mDdCBEs can be loosely defined as spanning nucleotides 4~11 downstream of a TALE-binding site.”

3. The use of 800 ng mRNA was efficient as transfection of plasmid DNA. How many cells do it transfect?

Response: We used 8×10^4 HEK293T cells/well in our transfection experiments. This point is indicated in the Methods section.

Reviewer #3 (Remarks to the Author):

In this study, the authors generated variants of DddAtox, the deaminase catalytic domains in the inter-bacterial toxin DddA from *Burkholderia cenocepacia*, by using structure-based site-directed mutagenesis and random mutagenesis. Some of these variants were fused to Cas9 nickase and TALE DNA-binding proteins for cytosine base editing in nuclear and mitochondrial DNA editing in human and mouse cells. The authors' approach enabled mitochondrial DNA editing using monomeric DdCBE, which is important progress in the field of mitochondrial genome editing. In addition, the authors also showed that mitochondrial genome-wide off-target effects can be reduced or eliminated by using mDdCBE-encoding mRNAs instead of plasmid DNA.

This well-designed study is timely. The authors achieved base editing using monomeric DdCBEs, which is an important contribution to the field of mitochondrial genome editing. I would recommend the publication of this exciting study after addressing only a minor point shown below.

Minor comment

1. The authors used E1347A for base editing, which is not compatible with the result of a previous study showing that the E1347A mutant is catalytically inactive (Mok et al. Nature 2020). Thus, this point should be addressed. I recommend testing the E1347A mutant under cell-free in vitro conditions whether it can induce cytosine deamination or not.

Response: We have now performed additional experiments and described the results on p.5 as follows: **“Indeed, we were able to confirm the deaminase activity of the recombinant E1347A-D10A nCas9 fusion protein, expressed in and purified from *E. coli*, under cell-free in vitro conditions using a PCR amplicon containing the TYRO3 site (Supplementary Fig. 1).”**

REVIEWERS' COMMENTS

Reviewer #1 (Remarks to the Author):

The manuscript is a revision and this reviewer also reviewed the previous version (reviewer #1).

Mok et al improved DddA-derived cytosine base editors (DdCBEs) by selecting mutations in DddA which make the protein less toxic. This allowed to use them as monomeric fusion to Cas9 or one TALE DNA-binding domain and changed the target cytosines which could be edited. This approach reduces the cloning effort and allows a highly flexible use of this novel genome editing tool.

The authors addressed most points that were raised by the three reviewers and included new experimental data. Practically all of the modifications are well done. I agree with the authors that the new data showing that monomeric mDdCBEs delivered by AAV to human cells are a very welcome and significant addition. Overall, the manuscript has well improved and it is a significant contribution which will raise broad interest.

I only have one comment:

lines 52-53: I insist that the sentence: "...because TALE proteins preferentially bind to target sites with a thymidine at both the 5' and 3'-termini..." is false. TALEs do never require a "T" at the 3' end. If the authors construct the TALEs in such a way that they do require a "T" they need to state that. The authors might choose to change the sentence accordingly, or just remove it. One suggestion would be: "...because TALE proteins constructed with our cloning kit preferentially bind to target sites with a thymidine at both the 5' and 3'-termini...". (The authors might actually find a better solution)

The modified sentence (lines 53-54): "..., although it is possible to design TALE proteins that recognize sites with no thymine at the 5' or 3' terminus, ..." is not compensating the first wrong half of the sentence. It still gives the impression that normally TALEs bind a T at the 3' end which they do not.

At a later point in the manuscript, the authors corrected the text very well (lines 189-193) and I agree that this statement now clarifies the issue to non-expert readers. In this context, I agree to keep Fig. 4 (and Fig. 4a) in the manuscript.

All other points which I raised are addressed well.

Reviewer #2 (Remarks to the Author):

In this revised manuscript, the authors well addressed most of points raised by the reviewers. The manuscript has been substantially improved. Nevertheless, I would like to see the following minor concern to be addressed in the manuscript before its publication in Nature Communication.

Minor concern

The authors described they achieved the editing efficiency as high as 99.1% using AAV at the ND4 site, which is nearly homoplasmic mutations (line 205). While in the abstract, they claimed mDdCBEs expressed via AAV in cultured human cells can achieve homoplasmic C-to-T editing (line 20). To keep consistency, it is better to add "nearly" before "homoplasmic".

Reviewer #3 (Remarks to the Author):

The authors have addressed my concern. I think that the manuscript is now suitable for publication.

POINT-BY-POINT RESPONSE

We thank three anonymous reviewers for constructive criticisms and thoughtful comments. We have now fully addressed the remaining concerns of the reviewers in our revised manuscript.

Reviewer #1 (Remarks to the Author):

The manuscript is a revision and this reviewer also reviewed the previous version (reviewer #1).

Mok et al improved DddA-derived cytosine base editors (DdCBEs) by selecting mutations in DddA which make the protein less toxic. This allowed to use them as monomeric fusion to Cas9 or one TALE DNA-binding domain and changed the target cytosines which could be edited. This approach reduces the cloning effort and allows a highly flexible use of this novel genome editing tool.

The authors addressed most points that were raised by the three reviewers and included new experimental data. Practically all of the modifications are well done. I agree with the authors that the new data showing that monomeric mDdCBEs delivered by AAV to human cells are a very welcome and significant addition. Overall, the manuscript has well improved and it is a significant contribution which will raise broad interest.

I only have one comment:

lines 52-53: I insist that the sentence: "...because TALE proteins preferentially bind to target sites with a thymidine at both the 5' and 3'-termini..." is false. TALEs do never require a "T" at the 3' end. If the authors construct the TALEs in such a way that they do require a "T" they need to state that. The authors might choose to change the sentence accordingly, or just remove it. One suggestion would be: "...because TALE proteins constructed with our cloning kit preferentially bind to target sites with a thymidine at both the 5' and 3'-termini...". (The authors might actually find a better solution)

The modified sentence (lines 53-54): "..., although it is possible to design TALE proteins that recognize sites with no thymine at the 5' or 3' terminus, ..." is not compensating the first wrong half of the sentence. It still gives the impression that normally TALEs bind a T at the 3' end which they do not.

Response: We have now modified the sentence, as follows: "..because TALE proteins constructed with our golden-gate cloning kit preferentially bind to target DNA sites with a thymidine at both the 5'- and 3'-termini, although it is possible to design TALE proteins that recognize sites with no thymidine at the 5'¹³,..."

At a later point in the manuscript, the authors corrected the text very well (lines 189-193) and I agree that this statement now clarifies the issue to non-expert readers. In this context, I agree to keep Fig. 4 (and Fig. 4a) in the manuscript.

All other points which I raised are addressed well.

Reviewer #2 (Remarks to the Author):

In this revised manuscript, the authors well addressed most of points raised by the reviewers. The manuscript has been substantially improved. Nevertheless, I would like to see the following minor concern to be addressed in the manuscript before its publication in Nature Communication.

Minor concern

The authors described they achieved the editing efficiency as high as 99.1% using AAV at the ND4 site, which is nearly homoplasmic mutations (line 205). While in the abstract, they claimed mDdCBEs expressed via AAV in cultured human cells can achieve homoplasmic C-to-T editing (line 20). To keep consistency, it is better to add “nearly” before “homoplasmic”.

Response: We have now added “nearly” in the sentence, as suggested by this reviewer.

Reviewer #3 (Remarks to the Author):

The authors have addressed my concern. I think that the manuscript is now suitable for publication.